EMBO
Molecular Medicine

# A single mutation in Taiwanese H6N1 influenza hemagglutinin switches binding to human-type receptors

Robert P de Vries[1,2,†], Netanel Tzarum[3,†], Wenjie Peng[1,†], Andrew J Thompson[1,†],
Iresha N Ambepitiya Wickramasinghe[4], Alba T Torrents de la Pena[5], Marielle J van Breemen[5],
Kim M Bouwman[4], Xueyong Zhu[3], Ryan McBride[1], Wenli Yu[3], Rogier W Sanders[5,6],
Monique H Verheije[4], Ian A Wilson[3,7,*] ID & James C Paulson[1,**] ID

## Abstract

In June 2013, the first case of human infection with an avian H6N1 virus was reported in a Taiwanese woman. Although this was a single non-fatal case, the virus continues to circulate in Taiwanese poultry. As with any emerging avian virus that infects humans, there is concern that acquisition of human-type receptor specificity could enable transmission in the human population. Despite mutations in the receptor-binding pocket of the human H6N1 isolate, it has retained avian-type (NeuAcα2-3Gal) receptor specificity. However, we show here that a single nucleotide substitution, resulting in a change from Gly to Asp at position 225 (G225D), completely switches specificity to human-type (NeuAcα2-6Gal) receptors. Significantly, G225D H6 loses binding to chicken trachea epithelium and is now able to bind to human tracheal tissue. Structural analysis reveals that Asp225 directly interacts with the penultimate Gal of the human-type receptor, stabilizing human receptor binding.

**Keywords** glycan array; hemagglutinin; influenza A virus; sialic acid; X-ray crystallography
**Subject Categories** Microbiology, Virology & Host Pathogen Interaction

## Introduction

2013 was a remarkable year for influenza A virus (IAV) zoonosis. Multiple avian virus subtypes successfully crossed the species barrier into humans including H5N1, H7N7, H7N9, H9N2, and H10N8 viruses (Freidl *et al*, 2014), as well as a single infection of a novel H6N1 virus in a 20-year-old Taiwanese woman (CDC, 2013; Shi *et al*, 2013; Wei *et al*, 2013). Fortunately, none of these viruses has so far acquired the ability to transmit efficiently between humans (Paulson & de Vries, 2013; Xu *et al*, 2013; Tzarum *et al*, 2015; Wang *et al*, 2015; Yang *et al*, 2015; Zhang *et al*, 2015). Previous human pandemics have largely been of avian origin and required a shift in receptor specificity from glycans with a sialic acid-linked α2-3 to galactose (avian-type receptor) to α2-6-linked sialosides (human-type receptor; Matrosovich *et al*, 2009). Thus, understanding the potential for new avian viruses to acquire human receptor specificity is an important factor in assessing potential pandemic threats.

Over the last 100 years, human influenza pandemics have been caused by only three influenza A virus subtypes, H1N1, H2N2, and H3N2. In each case, two amino acid changes in the HA were required to change the specificity of the avian virus progenitor to recognition of human-type receptors (Rogers & D'Souza, 1989; Matrosovich *et al*, 2000). H1N1 viruses were able to cross the species barrier through introduction of E190D and G225D mutations, while H2 and H3 viruses achieved receptor switching through Q226L and G228S (H3 amino acid numbering will be used throughout). Contemporary H3N2 viruses have now evolved to contain different amino acids at several positions in the receptor-binding site (RBS), yet retain preference for α2-6-linked sialosides (Fig 1A;

1 Departments of Molecular Medicine & Immunology and Microbiology, The Scripps Research Institute, La Jolla, CA, USA
2 Department of Chemical Biology and Drug Discovery, Utrecht Institute for Pharmaceutical Sciences, Utrecht University, Utrecht, The Netherlands
3 Department of Integrative Structural and Computational Biology, The Scripps Research Institute, La Jolla, CA, USA
4 Pathology Division, Department of Pathobiology, Faculty of Veterinary Medicine, Utrecht University, Utrecht, The Netherlands
5 Department of Medical Microbiology, Academic Medical Center, University of Amsterdam, Amsterdam, The Netherlands
6 Department of Microbiology and Immunology, Weil Medical College of Cornell University, New York, NY, USA
7 Skaggs Institute for Chemical Biology, The Scripps Research Institute, La Jolla, CA, USA
*Corresponding author. Tel: +1 858 784 9706; E-mail: wilson@scripps.edu
**Corresponding author. Tel: +1 858 784 9634; E-mail: jpaulson@scripps.edu
†These authors contributed equally to this work

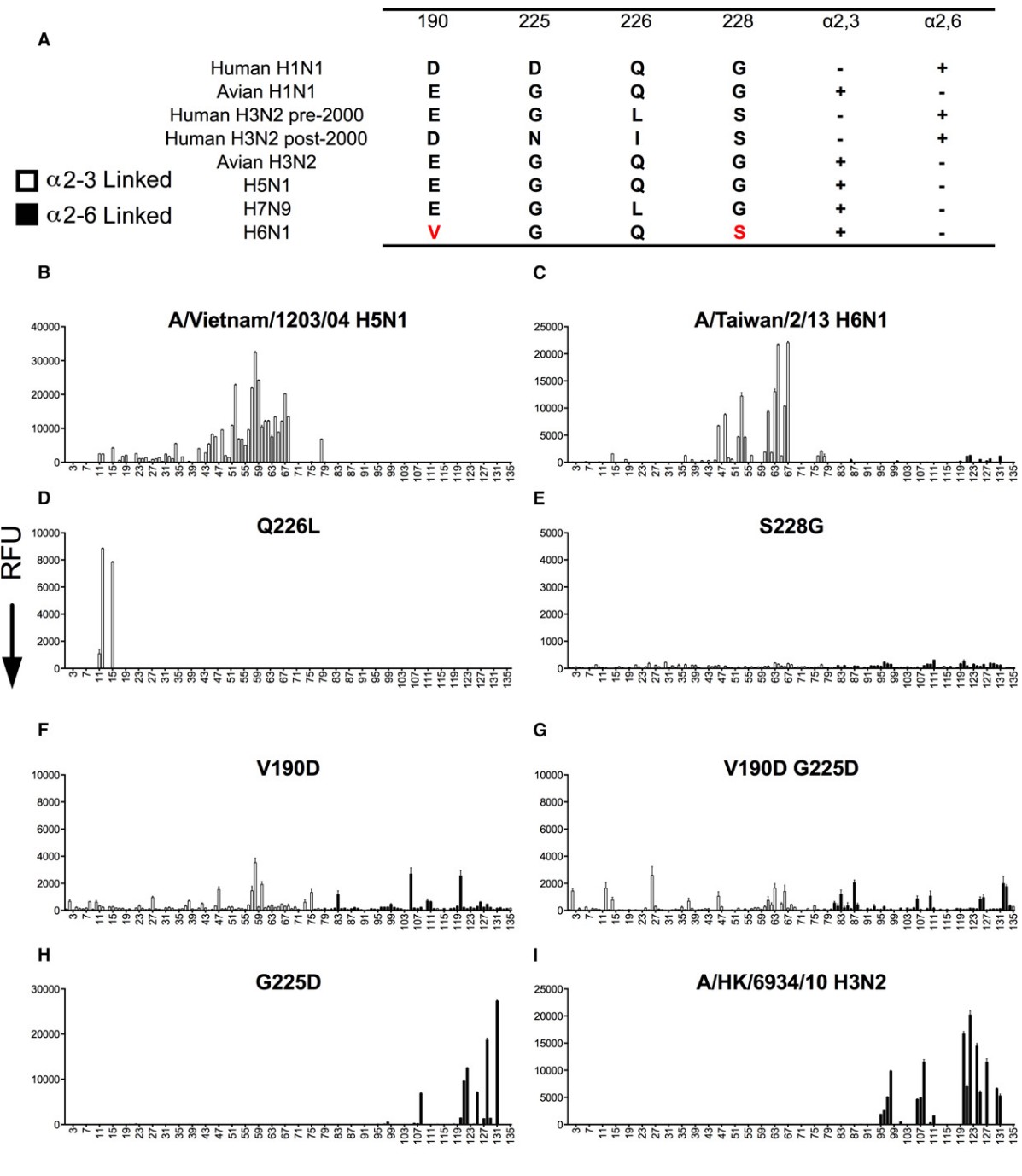

**Figure 1. Receptor binding analyses of recombinant H5, H6, and H3 HAs using an expanded sialoside glycan array.**

A Amino acids that confer human and avian receptor specificity in H1N1 and H3N2 viruses. H5N1 and H7N9 are other zoonotic viruses with avian receptor specificity. Binding to either avian (α2,3) or human-type (α2,6) receptors is indicated by either + or −.

B A/Vietnam/1203/04 H5N1 HA only binds to α2-3-linked sialic acids (*n* = 3).

C H6N1 wild-type HA binds to a subset of α2-3 sialosides.

D H6 Q226L HA binds to α2-3-linked sialic acids with a sulphated LacNAc.

E–G S228G, V190D single mutants, and V190D G225D double mutants hardly bind to any glycans on the array.

H, I (H) H6 G225D mutant binds solely to human-type receptors, which are also bound by (I) a human seasonal H3N2 control HA, A/Hong Kong/6934/10.

Data information: All HA proteins are analyzed on three independent arrays with different batches of protein (*n* = 3). The mean signal and standard error were calculated from four of the six independent replicates on the array after omitting the high and low values. α2-3-linked sialosides in white bars (glycans 11–79 on the *x*-axis) and α2-6-linked sialosides in black (glycans 80–135). Glycans 1–10 are non-sialylated controls (see also Appendix Table S1).

Matrosovich *et al*, 2000; Lin *et al*, 2012; Peng *et al*, 2017). Analysis of H6 HA sequences from poultry isolates and the recent human H6N1 isolate reveals a combination of four amino acids at these positions that are not seen together in other avian viruses, namely V190, G225, Q226, and S228. While G225 and Q226 are present in other avian viruses, V190 and S228 are non-canonical residues for avian influenza viruses. In addition, Wang *et al* (2015) observed that the human isolate (A/Taiwan/2/2013) also contained a P186L mutation that was distinct from recent Taiwanese H6N1 poultry isolates. Careful analysis of the binding avidity of H6 hemagglutinins (HA) to a pair of sialoglycan receptor analogs (NeuAcα2-3/6Galβ1-4GlcNAcβ1-3Galβ1-4GlcNAcβ-biotin) using surface plasmon resonance (SPR) showed that the poultry and human virus H6 HAs both bound weakly to the human-type receptor analog (α2-6 linked). In contrast, while the chicken H6 HA bound much more avidly to the avian receptor analog (α2-3 linked), the human H6 isolate, containing a P186L mutation, bound with significantly reduced avidity, resulting in a slight binding preference for the human receptor analog in this particular assay. Subsequent reports evaluating receptor specificity in greater depth using glycan microarrays with a diverse library of human-type and avian-type receptor glycans showed that the human H6N1 HA retained avian-type receptor specificity, with negligible binding to human-type receptors (Tzarum *et al*, 2015; Yang *et al*, 2015).

Through systematic introduction of mutations responsible for receptor specificity switching in other human influenza viruses, we found that a single G225D mutation in the human H6 HA strongly shifted receptor specificity from avian to human type. This was evident both in glycan microarrays and in binding to chicken and human airway epithelium. Other mutations examined had no comparable shift in receptor specificity. Because increased stability of the HA is also a factor in transmission between humans and ferrets, we analyzed the thermostability of the mutants by differential scanning calorimetry. Thermostability analysis of the H6 G225D variant actually revealed a decrease in melting temperature ($T_m$) compared to wild-type H6N1 and thus has a thermostability profile comparable to other avian influenza viruses including H5N1 (Linster *et al*, 2014). While it is not known if the G225D change alone would be sufficient to impact transmission in mammals, it would be prudent to monitor for such mutations in routine surveillance of H6N1 in Taiwanese poultry (Wang *et al*, 2015).

# Results

## Mutational analyses and receptor binding of established specificity mutations

Although the human H6N1 isolate (A/Taiwan/2013) has shown decreased avian-type receptor specificity in some assays (Wang *et al*, 2015), it has maintained strong avian-type receptor specificity relative to other avian and human viruses (Tzarum *et al*, 2015; Yang *et al*, 2015). To investigate the potential for human H6 HA to acquire human-type receptor specificity, we employed site-directed mutagenesis of the RBS (Fig 1A) and produced the respective soluble recombinant trimeric HA proteins in HEK293S GnTI$^{-/-}$ cells (de Vries *et al*, 2010). Receptor specificity was assessed on a glycan microarray that also contain linear and branched *O*- and *N*-linked glycans with extended poly-*N*-acetyl-lactosamine chains, which were found to be the preferred receptors of the 2009 H1N1 pandemic virus (A/CA/04/09) and recent human H3N2 viruses (see Appendix Table S1 for complete list; Peng *et al*, 2017). As observed for a reference H5N1 avian virus HA, the wild-type human H6 HA binds solely to avian-type receptors, but with remarkable specificity toward extended *N*-linked glycans (#53–67) (Fig 1B and C). To assess the receptor-switching potential of mutations known to confer human-type specificity in H1N1, H2N2, and H3N2 viruses, we introduced similar changes within A/Taiwan/2013 H6 HA at positions 190, 225, 226, and 228. Interestingly, Q226L resulted in a general loss of binding to avian-type receptors, but gained specificity for NeuAcα2-3Gal receptors where the antepenultimate GlcNAc is 6-sulphated (Fig 1D), an observation consistent with reports for certain other avian virus strains (Gambaryan *et al*, 2008; Peacock *et al*, 2017). H6 single mutants containing S228G and V190D, as well as a V190D-G225D double mutant, resulted in significant decreases in receptor binding with no clear specificity for either human or avian receptors (Fig 1E–G). Surprisingly, however, introduction of the single G225D mutation conferred significant binding and strong specificity for human-type receptors (Fig 1H). Binding was observed only to α2-6-linked sialosides, with preference for selected extended *N*-linked glycans similar to a recent human H3N2 HA (Fig 1I, Appendix Table S2). Notably, G225D showed no detectable binding to the α2-3-linked (#18) or α2-6-linked (#84) sialyl-di-LacNAc structures frequently used as an exemplary receptor pair to characterize receptor specificity in solid-phase assays (Chandrasekaran *et al*, 2008; Wang *et al*, 2015). Wild-type and G225D mutant H6 HAs produced in insect cells also exhibited preferential specificity for avian- and human-type receptors, respectively. However, binding appeared to be of higher avidity, presumably due to the smaller glycans attached by these cells (Appendix Fig S1).

## Further analyses of amino acids that may contribute to human-type receptor binding

Since G225D is a hallmark mutation in H1N1 viruses that confers binding to human-type receptors, we tested several other amino acid positions known to influence receptor binding in H1 and H3 viruses, including A222K, R227A, L186S, and the L186P mutation that was previously shown to increase avidity of the chicken H6 HA for avian-type receptors (see Fig 2A; Wang *et al*, 2015). A222K and R227A single mutants did not affect receptor-binding properties in this H6 HA (Fig 2B and C, left), but further addition of G225D into these single amino acid mutant backgrounds both showed an increase in binding to human-type receptors, with the A222K/G225D exhibiting a complete switch in specificity (Fig 2B and C, right). Leucine at 186 in the Taiwanese H6 human isolate is unique because it is not found in human or avian isolates. Single mutations at this position to the more commonly observed 186S and 186P variants show reduced binding to most avian-type receptors, with binding increases restricted only to short α2-3-linked glycans containing 6S-GlcNAc (Fig 2D and E, left). The L186S/P-G225D double mutants showed weak human-type receptor specificity (Fig 2D and E, right), consistent with the impact of G225D alone (Fig 1H). The results taken together show that the G225D mutation can be a significant determinant for human-type receptor binding in A/Taiwan/2/13 H6 HA and its impact on receptor specificity is tolerant of further RBS mutations.

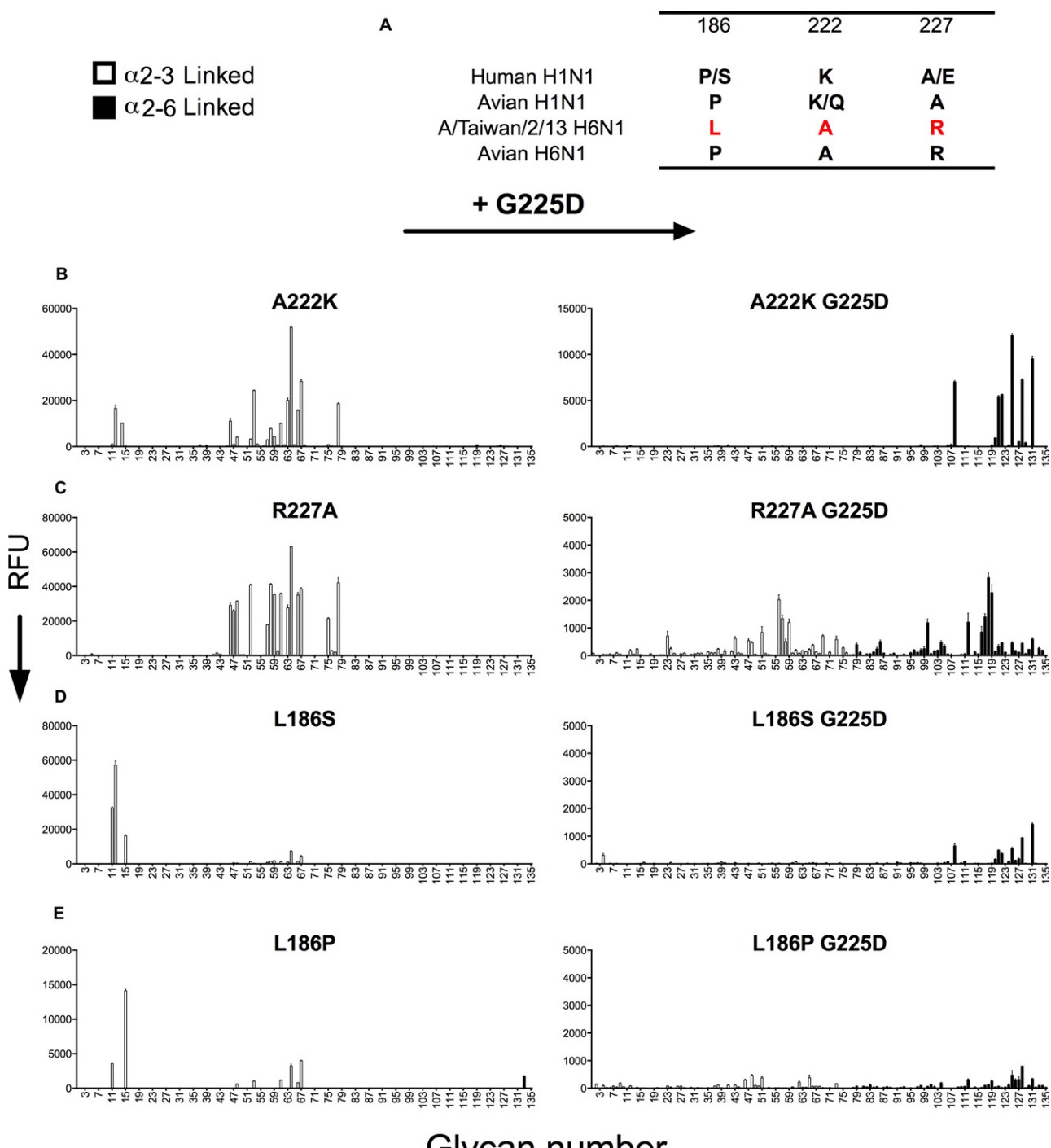

**Figure 2.  Receptor binding analysis of additional H6N1 mutant HAs.**

A   Residues that are important for receptor binding in H1N1 viruses differ substantially in H6N1.

B   A222K does not affect receptor binding, but addition of the G225 mutation to A222K confers specificity for human-type receptors.

C   R227K does not affect specificity for avian-type receptors, but addition of G225D to R227K confers some binding to α2-6-linked sialosides.

D   The H6 L186S mutant binds preferentially to α2-3-linked sialosides with a sulphated LacNAc. The addition of G225D in this background results in loss of binding to avian-type and weak binding to human-type receptors.

E   The L186P mutation results in similar specificity to L186S and addition of G225D results in a loss of binding to the array.

Data information: All HA proteins are analyzed on three independent arrays with different batches of protein (*n* = 3). The mean signal and standard error were calculated from four of the six independent replicates on the array after omitting the high and low values. α2-3-linked sialosides in white bars (glycans 11–79 on the *x*-axis) and α2-6-linked sialosides in black (glycans 80–135). Glycans 1–10 are non-sialylated controls (see also Appendix Table S1).

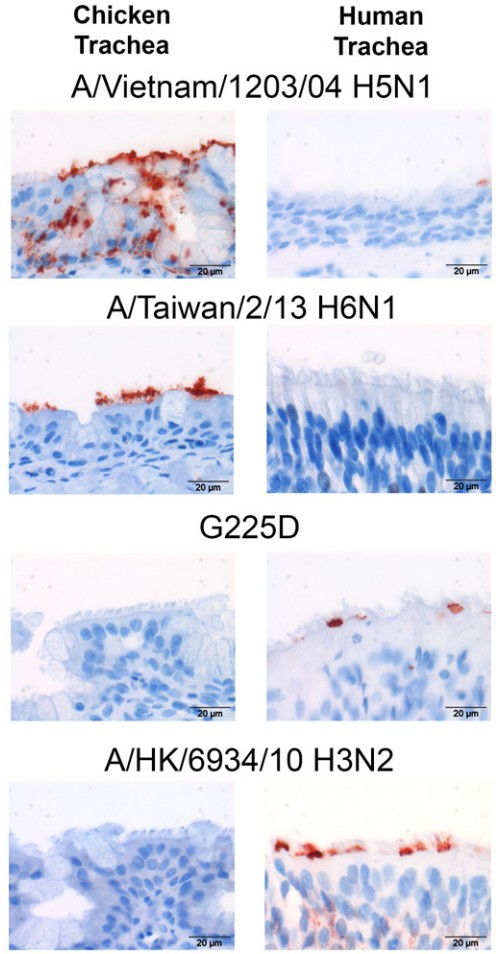

**Figure 3. Analysis of mutant human H6N1 HAs binding to chicken and human trachea epithelial cells.**

Binding of recombinant HA proteins A/Vietnam/1203/04 H5N1, A/Taiwan/2/13 H6N1, H6N1 G225D mutant, and a human seasonal H3N2 control (A/Hong Kong/6934/10) to chicken and human tracheal tissue. The images are representative of three independent assays (*n* = 3). Binding is detected using by HA antibody complexes containing an anti-StrepTag-HRP and a goat anti-mouse HRP and developed with AEC.

## Binding to chicken and human respiratory tissues

Binding to human respiratory epithelium is an important factor for zoonotic viruses to be able to successfully transmit between humans (van Riel *et al*, 2007; de Graaf & Fouchier, 2014). Thus, we tested the ability of the H6 wild type and G225D single mutant to bind to airway epithelial cells in chicken and human tracheal tissue sections. After quality control using plant lectins SNA and MAL to specifically detect α2-6 and α2-3 sialosides (Appendix Fig S2), we used the same control HAs employed in Fig 1 to stain the respiratory tissue. Similar to the avian reference strain A/Vietnam/1203/04 (H5N1), wild-type H6 HA binds epithelia sections of chicken trachea, but not human trachea (Fig 3). In contrast, the G225D mutant has acquired the ability to bind human trachea epithelium and lost binding to chicken trachea. As expected, the human seasonal H3 HA control stains only human but not chicken trachea.

## G225D leads to typical human-type receptor binding as observed at the atomic level

To examine the structural features that underlie the specificity switch of the H6 HA G225D mutant, we determined its crystal structure in both apo and complex form with human and avian receptor analogs (Appendix Table S3). The G225D H6 mutant structure at 2.19 Å is highly similar to wild-type H6 HA [PDB entry 4XKD, Cα root-mean-squares deviation (RMSD) of 0.13 and 0.14 Å to the HA monomer and the RBS subdomain (amino acid 117–265), respectively], with only slight differences due to changes in the rotamers of side chains of RBS N137, L186, and Q226 (Fig 4A). In the structure of the complex, the human-type receptor analog, 6′-SLN, binds in a *cis* conformation, extending into the space between the 190-helix and 220-loop (Fig 4B and C, and Appendix Fig S3), that is similar to the minimum energy solution conformation of α2-6 sialosides (Sabesan *et al*, 1991; Eisen *et al*, 1997). 6′-SLN forms hydrogen bonds with the main-chain carbonyl and the side chain of D225 through the 4-hydroxyl and 3-hydroxyl of Gal-2, respectively, and with the Q226 side chain via the 4-hydroxyl of Gal-2, that stabilize the *cis* conformation. The phi angle between Sia-1 and Gal-2 ($O_6$Sia-$C_2$Sia-O-$C_6$Gal) is similar to human analogs with other avian and humans HAs (Appendix Fig S4; Xu *et al*, 2012), but different from the H6 HA wild type with 6′-SLN (Tzarum *et al*, 2015; phi of 70° for the G225D mutant and ~120° for wild-type HA, Fig 4D). A similar phenotype of change in the conformation of α2-6 sialoside receptors, as a consequence of the change in the receptor specificity of avian-origin HAs, has been shown for the airborne transmissible H5N1 HA and the A/Jiangxi/IPB13a/2013 H10N8 (Xiong *et al*, 2013; Zhang *et al*, 2013; Tzarum *et al*, 2017).

To understand the differential binding to avian-type receptors, crystal structures of complexes of the G225D mutant with a 50-fold excess of trisaccharide avian receptor analogs 3′-SLN

**Figure 4. Crystal structure of the H6 HA G225D mutant in complex with a human receptor analog 6′-SLN.**

A   Structural comparison of the RBS of the G225D mutant HA (gray) and the wild-type HA (green) displaying slight conformation changes in the vicinity of the mutation site Asp225. The conserved secondary elements of the HA RBS (130-loop, 190-helix, and 220-loop) are labeled and shown in cartoon representation. Selected residues and the receptor analogs are labeled and shown in sticks.

B   The glycan structure of human receptor 6′-SLN, Neu5Acα2-6Galβ1-4GlcNAc. Purple diamond, *N*-acetylneuraminic acid; yellow circle, galactose; blue square, *N*-acetylglucosamine.

C   Hydrogen bond interactions of the H6 G225D mutant RBS with Gal-2 of human receptor analog 6′-SLN. The receptor analog is labeled, with carbons colored in yellow and shown in sticks. Sia is abbreviation for sialic acid, Gal for galactose, and GlcNAc for *N*-acetylglucosamine.

D   Superposition of 6′-SLN receptor analog from H6 G225D mutant complex (gray) compared to the H6 wild-type complex (green) indicates conformational changes arising from rotation around the linkage between Sia-1 and Gal-2 (phi changes from 120° for wild type to 70° in mutant structures). The 6′-SLN receptor analog and the RBS Asp225 and Gln226 are labeled and shown in sticks.

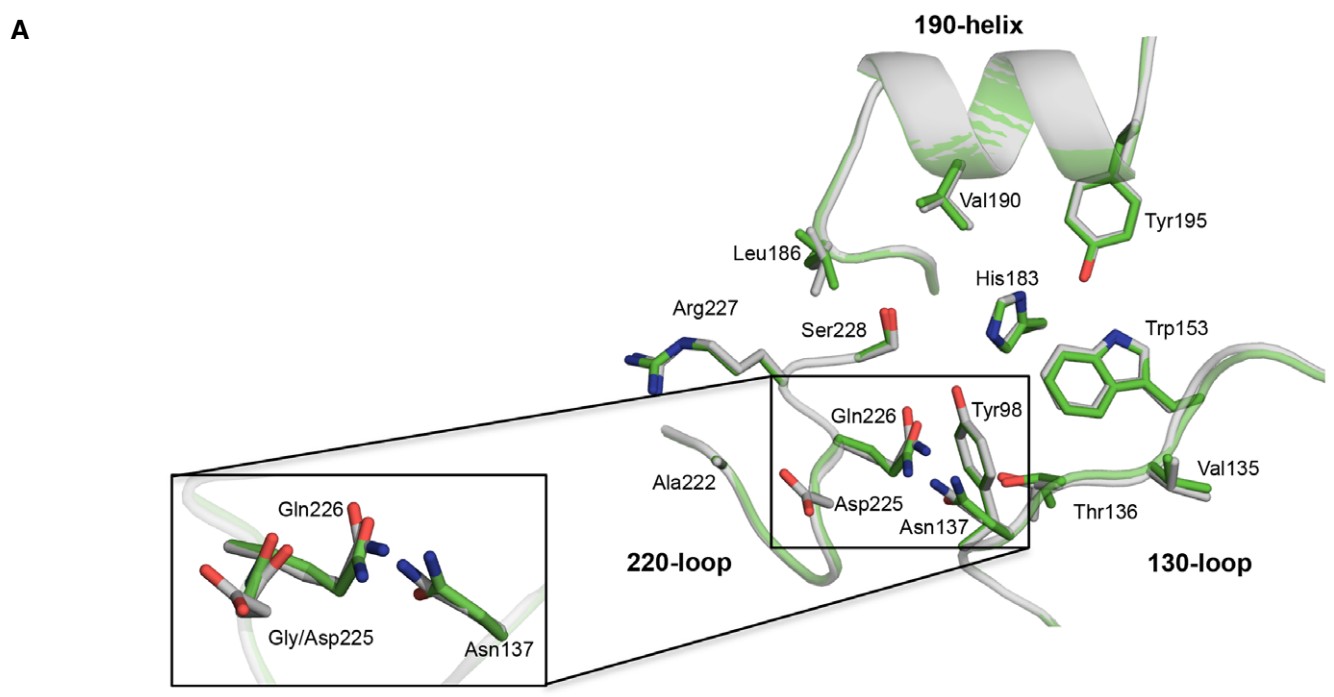

**A**

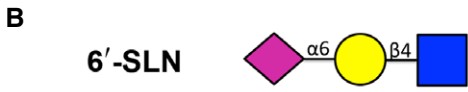

**B**

6′-SLN

**C**

H6 _G225D_ HA – 6′-SLN complex

**D**

6′-SLN _H6 wt_ - 6′-SLN _H6 G225D_

**Figure 4.**

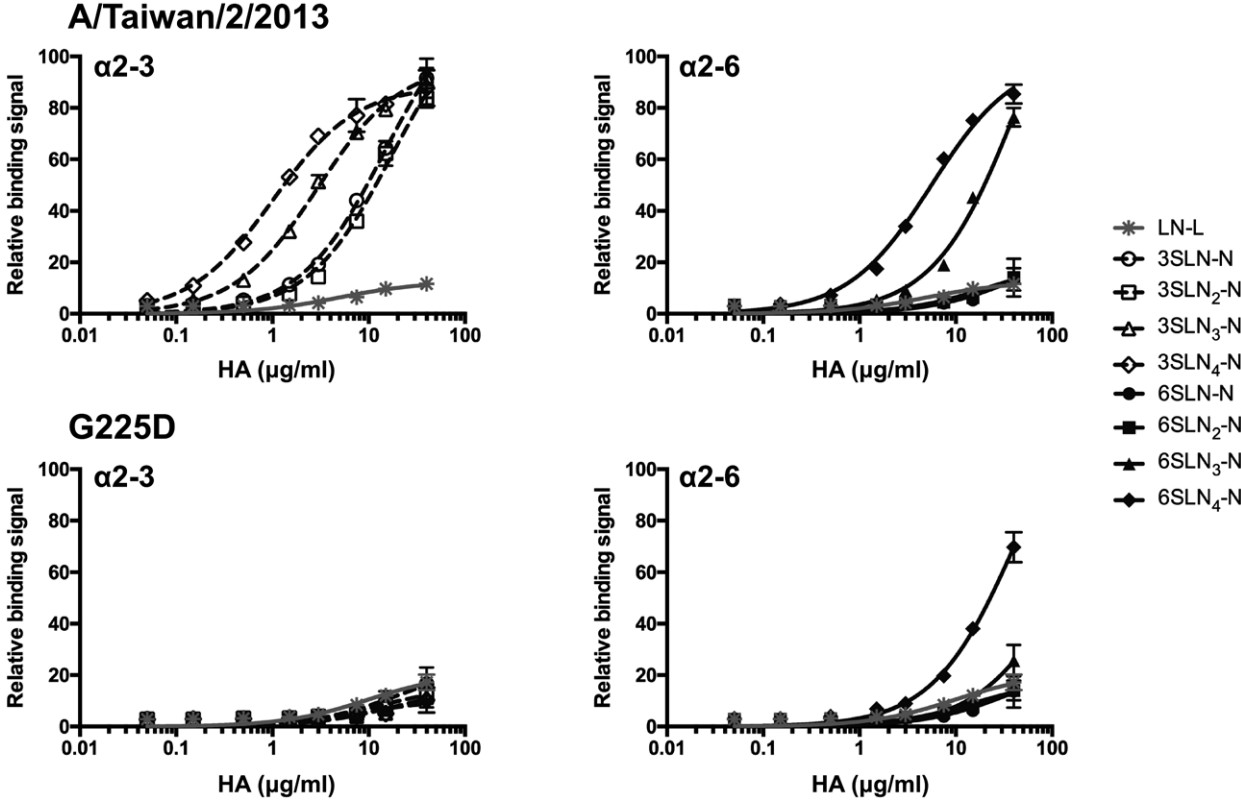

**Figure 5.   Binding of A/Taiwan/2/13 and the G225D mutant HA to biotinylated *N*-glycan receptors, as determined by glycan ELISA.**
Wild-type HA (upper panels) binds strongly to avian-type receptors terminating in α2-3 sialic acid (left, white open shapes) with weaker binding to human-type (α2-6) receptors (right, black closed shapes). Despite lower avidity, G225D (lower panels) shows strong human specificity, with slightly reduced binding to α2-6 *N*-glycans and no detectable interaction with avian receptors. Assays are conducted with biantennary, *N*-linked glycans (N) with one to four LacNAc (LN, Galβ1-4GlcNAc) repeats terminated with sialic acid (S) in α2-3 or α2-6 linkage (SLN$_{1-4}$-N) (*n* = 2). An asialo, mono-LacNAc (LacNAc-biotin, LN-L) was used as a negative binding control.

(NeuAcα2-3Galβ1-4GlcNAc) and with pentasaccharide LSTa (NeuAcα2-3Galβ1-3GlcNAcβ1-3Galβ1-4Glc) (Appendix Fig S5A) were determined at 2.9 Å and 2.1 Å, respectively, where three and two glycan monosaccharides could be visualized (Appendix Fig S5B and D, and Appendix Table S3). 3′-SLN binds in a cis conformation similar to its interaction in the wild-type HA with slight changes in Gal-2 and GlcNAc-3 presumably to prevent a steric clash between the Gal-2 6-hydroxyl and the aliphatic part of Asp225 (Appendix Fig S5C and E). Consequently, the distance between the Asp225 main-chain carbonyl, the Gal-2 6-hydroxyl, and the GlcNAc-3 3-hydroxyl is increased (from 2.6 Å and 3.2 Å, respectively, in the H6 HA wild-type complex, to 4.6 Å and 3.8 Å in the mutant complex), reducing the hydrogen bond interactions between Gal-2 and GlcNac-3 of 3′-SLN with the G225D mutant RBS. For LSTa, only Sia-1 and Gal-2 displayed electron density, implying very weak interactions between LSTa and the RBS. Similar to the wild-type complex, LSTa binds in a *cis* conformation forming hydrogen bond interactions between Gal-2 O6 and the carboxyl group of Asp225 (Appendix Fig S5D and E). We therefore conclude that the single amino acid mutation, G225D, in HA of H6N1 enables interaction with human-type receptors with a similar binding mode compared to other human HAs despite not having all of the canonical residues in human H1 (Val190 instead of Asp190) or H2/H3 (Gln228 instead of Ser228) HAs (Figs 1 and 4).

**Relative avidity of H6 HAs to *N*-linked glycans**

To quantify the binding avidities of A/Taiwan/2/13 and the G225D mutant HAs and assess in detail the relative magnitude of the specificity switch to human-type receptor binding, we conducted a glycan ELISA using a series of biantennary *N*-linked glycans featuring either terminal NeuAcα2-3Gal or NeuAcα2-6Gal (Fig 5). These glycans are consistently observed as preferred receptors on the glycan array (Peng *et al*, 2017). A/Taiwan/2/13 was selective for avian-type receptors with some detectable binding to human-type receptors and showed little preference for glycan length, consistent with the glycan array results. The G225D mutant lost all binding to avian-type receptors and now specifically binds to human-type receptors. Interestingly, the apparent avidity to biantennary *N*-glycans with 3 or 4 LacNAc repeats was significantly decreased, even lower compared to the wild-type protein ($K_d$ values are given in Appendix Table S4). However, G225D only recognized human-type receptors and can therefore be considered as highly specific. We also tried to analyze the avidity of the H6 wild type and the G225D mutant on PAA-conjugated sialyl-di-LacNAc in an ELISA-like assay (McBride *et al*, 2016); however, both HAs failed to bind these receptors, as observed previously for H3 HA proteins (Peng *et al*, 2017). Both H6 proteins also failed to hemagglutinate erythrocytes of different species. We thus conclude that both H6 proteins are

                    

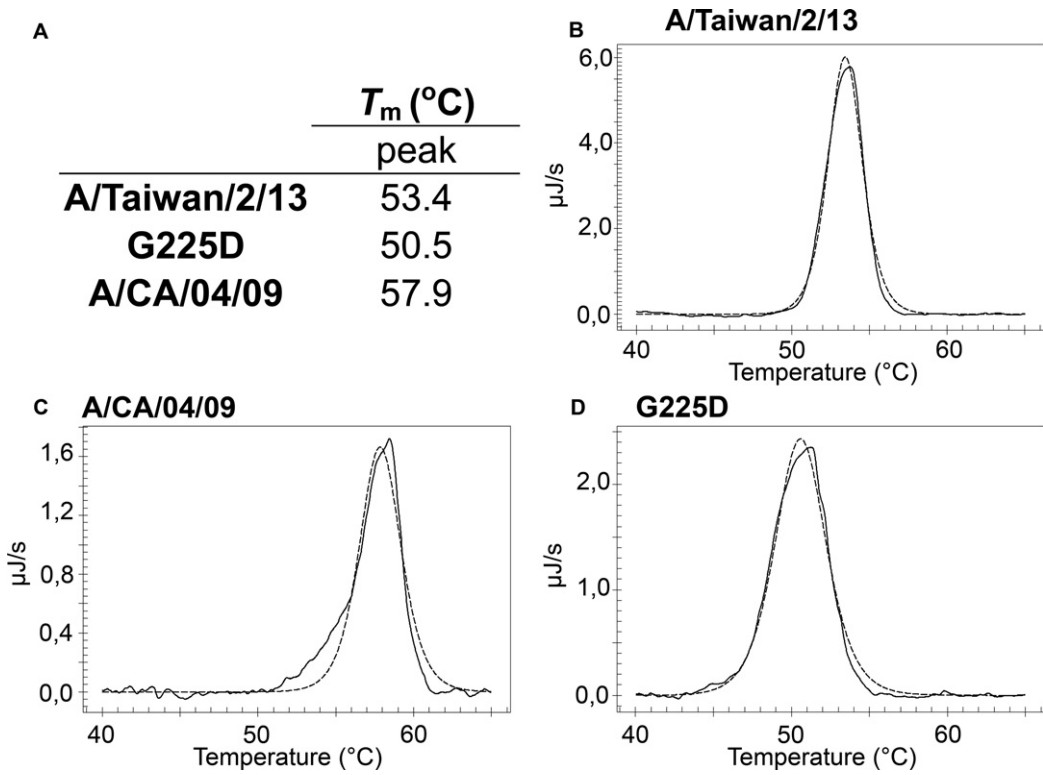

**Figure 6. Melting curves of recombinant HA by DSC to determine the thermostability of A/Taiwan/2/13, G225D, and A/CA/04/09.**

A    Summary of the DSC experiments listing the $T_m$ values for each recombinant HA.

B–D    Individual melting curves for A/Taiwan/2/13 (B), A/CA/04/09 (C) and G225D (D). The raw data are depicted in solid lines, while the fitted curves, from which the $T_m$ values were derived, are depicted with a dotted line. All proteins were analyzed by DSC twice (*n* = 2).

highly specific for complex branched *N*-glycans and that the G225D converts binding to human-type receptors, with a highly similar receptor binding profile compared to contemporary human H3N2 influenza viruses.

### G225D reduces H6 HA stability

The stability of HA in acidic environments is a determinant for airborne transmissibility of influenza viruses (Linster *et al*, 2014). In general, human virus HAs exhibit increased stability and fuse at lower pH than avian virus HAs. Thermal stability of an HA correlates with fusion at lower pH and can be used as an alternative measure of HA stability. Here, we analyzed the thermostability of H6, H6-G225D, and A/CA/04/09 HAs by differential scanning calorimetry (DSC; Fig 6A). The H6 protein had a midpoint of thermal denaturation ($T_m$) of 53.4 (Fig 6B). In comparison, the H1 HA of A/CA/04/09 has a $T_m$ of 57.9°C similar to that measured with whole human viruses and characteristic of an early 2009 pandemic human virus HA with fusion at pH 5.8 or lower (Fig 6C; Peacock *et al*, 2017; Russier *et al*, 2017). The H6 G225D protein had a midpoint of thermal denaturation ($T_m$) of 50.5°C (Fig 6D), showing that the G225D mutation if anything slightly decreases stability (Linster *et al*, 2014; de Vries *et al*, 2014). Thus, while the H6 G225D mutant adapts the HA to human receptor specificity, additional mutations to stabilize the HA for fusion at lower pH would likely be

required to support transmission by respiratory droplets between ferrets and between humans.

## Discussion

Although the H6N1 virus remains prevalent in Taiwanese poultry, only one infection has been reported in humans, in contrast to numerous infections from avian H5N1 and H7N9 associated with human exposure to poultry (Kandun *et al*, 2006; Murhekar *et al*, 2013). To gain insight into the basis for this human infection, several reports have evaluated the impact of mutations on receptor specificity in the HA of a human H6 isolate. Wang *et al* (2015) compared sequences from the human H6N1 isolate and H6N1 viruses from Taiwanese poultry from 1972 to 2005. Residues 186L, 190V, and 228S were found to decrease avidity to avian-type receptors, resulting in increased preference for human-type receptors in assays that employ fragments of avian and human-type receptors (Chandrasekaran *et al*, 2008; Wang *et al*, 2015; McBride *et al*, 2016). In contrast, analysis using glycan microarrays showed that the virus retained avian-type specificity with no human-type receptor binding (Tzarum *et al*, 2015; Yang *et al*, 2015). This apparent discrepancy is based on the glycans and assays used in the analysis. Although glycan arrays are also solid-phase assays, they employ a diverse set of glycans including complex branched glycan structures

found in airway tissues that are preferentially bound by human influenza viruses (Walther *et al*, 2013; de Graaf & Fouchier, 2014; Peng *et al*, 2017). Despite the low avidity of the glycans used in the SPR and microtiter assays, the increased preference for human-type receptors detected for the human H6 virus may be biologically relevant in infection. However, the preference of the human H6 virus for avian-type receptors in glycan arrays is clear (Fig 1C and Appendix Fig S1) and is further documented by binding of the HA to epithelial cells of chicken trachea, but not to human trachea (Fig 3).

Surprisingly, we found that the H6 HA has the capacity to completely switch to human-type receptor specificity with a single nucleotide change, resulting in the G225D mutation previously documented as one of two amino acid changes that confer switch in receptor specificity in human H1N1 viruses. Single amino acid changes affecting receptor binding can occur in a single infection as documented for a switch from human- to avian-type specificity in an H3N2 virus from a single passage in eggs (Rogers *et al*, 1985) and from avian- to transmissible human-type specificity for an H1N1 virus after a single passage in ferrets (Lakdawala *et al*, 2015). However, for H5N1, it has been shown that several mutations are required to attain human-type receptor specificity and respiratory droplet transmission between ferrets (Chen *et al*, 2012; Herfst *et al*, 2012; Imai *et al*, 2012).

Although it would be optimal to determine whether the G225D switch to human-type receptor specificity also supports aerosol droplet transmission in ferrets, such experiments comprise gain-of-function studies that are currently under a moratorium. We would like to emphasize that the mutant H6 was not placed into a replication-competent virus and studies were restricted to *in vitro* analysis of single viral proteins. In fact, while acquisition of human-type receptor specificity is believed to be a prerequisite for transmission of human influenza viruses, it is well known that an influenza polymerase E627K mutation is also often required for optimal infection (Gabriel *et al*, 2013). E627K is not present in avian and human H6N1 isolates but is detected in canine isolates (Lin *et al*, 2015). Furthermore, the H6 wild type and the G225D mutant demonstrate low thermostability that correlates to a high pH threshold for fusion, and are thus not adapted for human-to-human transmission (Linster *et al*, 2014). Also other as yet to be identified properties may need to be optimized for adaptation of the virus for transmission in man. Thus, our findings indicate that H6N1 infections should be closely monitored for any sign of change in receptor specificity or other changes that could lead to possible gain in human transmissibility.

# Materials and Methods

### Expression and purification of H1, H3, H5 and H6 HAs

Codon-optimized H3, H5, and H6 encoding cDNAs (Genscript, USA) of A/California/04/09 (Accession; FJ966082.1), A/Hong Kong/6934/10 (Accession; AJK01277.1), A/Vietnam/1203/04 (Accession; EF541403), and A/Taiwan/2/13 (Accession; EPI459855) were cloned into the pCD5 expression vector. All mutations were introduced using the Stratagene mutagenesis kit. The HA proteins were expressed in HEK293S GnTI$^{-/-}$ cells and purified from the cell culture supernatants as described previously (de Vries *et al*, 2010).

### Glycan microarray analysis of HAs

Purified, soluble trimeric HA was pre-complexed with horseradish peroxidase (HRP)-linked anti-strep-tag mouse antibody and with Alexa647-linked anti-mouse IgG (4:2:1 molar ratio) prior to incubation for 15 min on ice in 100 μl PBS-T and incubated on the array surface in a humidified chamber for 90 min. Slides were subsequently washed by successive rinses with PBS-T, PBS, and deionized $H_2O$. Washed arrays were dried by centrifugation and immediately scanned for fluorescence on a PerkinElmer ProScanArray Express confocal microarray scanner. Fluorescent signal intensity was measured using Imagene (Biodiscovery), and mean intensity minus mean background was calculated and graphed using MS Excel. For each glycan, the mean signal intensity was calculated from six replicates spots. The highest and lowest signals of the six replicates were removed and the remaining four replicates used to calculate the mean signal, standard deviation (SD), and standard error measurement (SEM). Bar graphs represent the averaged mean signal minus background for each glycan sample, and error bars represent the SEM values. A list of glycans on the microarray is included in Appendix Table S1.

### Tissue staining

Formalin-fixed, paraffin-embedded tissue sections were rehydrated in series of alcohol from 100, 96, and 70% and lastly distilled water. Endogenous peroxidase activity was blocked with 1% hydrogen peroxide for 30 min at room temperature. Tissue slides were boiled in citrate buffer pH 6.0 for 10 min at 900 kW in a microwave for antigen retrieval and washed in PBS-T for three times. Tissues were blocked with 3% BSA in PBS-T overnight at 4°C. HAs were pre-complexed with mouse anti-strep-tag- HRP antibodies (IBA) and goat anti-mouse IgG HRP antibodies (Life Biosciences) in a molar ratio of 4:2:1 in PBS-T with 3% BSA and incubated on ice for 15 min. After draining the slide, the pre-complexed HA was applied onto tissues and incubated for 90 min at RT. Sections were then washed in PBS-T, incubated with 3-amino-9-ethyl-carbazole (AEC; Sigma-Aldrich) for 15 min, counterstained with hematoxylin, and mounted with Aquatex (Merck). Images were taken using a charge-coupled device (CCD) camera and an Olympus BX41 microscope linked to CellB imaging software (Soft Imaging Solutions GmbH, Münster, Germany).

### Expression and purification of the H6 HA G225D for crystallization

The H6 HA cDNA of H6N1 A/Taiwan/2/13 [Global Initiative on Sharing All Influenza Data (GISAID) isolate ID: EPI_ISL_143275] was synthesized by Life Technologies (USA) and cloned into a pFastBac vector. H6 G226D HA was expressed in Hi5 insect cells with an N-terminal gp67 signal peptide, a C-terminal thrombin cleavage site, a foldon trimerization sequence, and a His$_6$-tag as described previously (Stevens *et al*, 2006). The expressed HA0 was purified via a His-tag affinity purification, dialyzed against 20 mM Tris–HCl pH 8.0, 100 mM NaCl, and then cleaved by trypsin (New England Biolabs, Ipswich, Massachusetts) to produce uniformly cleaved (HA1/HA2) and to remove the trimerization domain and His$_6$-tag. The digested protein was purified further by gel filtration chromatography using a Superdex-200 column (Pharmacia). The HA protein eluted as a trimer and was concentrated to 5 mg/ml.

## Crystallization and structural determination of H6 G225D HA

Crystals of the H6 G225D HA were obtained using the vapor diffusion sitting drop method (drop size 4 μl) at 20°C against a reservoir solution containing 10 mM NiCl$_2$, 0.1 M Tris pH 8.5, 20% (w/v) MPEG 2000, and 20% glycerol. Complexes of the HA with receptor analogs were obtained by soaking HA crystals in the reservoir solution that contained glycan ligands in a final concentration of 5 mM. Prior to data collection, the crystals were flash cooled in liquid nitrogen. Diffraction data were collected at the Advanced Photon Source (APS) and at the Stanford Synchrotron Radiation Lightsource (SSRL). Data were integrated and scaled using HKL2000 (Otwinowski & Minor, 1997). The initial H6 G225D apo structure was solved by molecular replacement using Phaser (McCoy *et al*, 2005) with H6N1 wild-type apo structure (PDB entry 4XKD) as the search model. The H6 G225D HA apo structure was then used as the starting model for structure determination of the H6 G225D HA-glycan complex structures. Structure refinement was carried out in Phenix (Adams *et al*, 2002) and model building with COOT (Emsley & Cowtan, 2004). Final data collection and refinement statistics are summarized in Appendix Table S3.

## Expression and purification of H6 from baculovirus expression system for glycan microarray analyses

The HA expression in baculovirus expression system was similar to the H6 HA expression for crystallization experiments. The expressed HA0s were purified through a His-tag affinity purification step and dialyzed against 20 mM Tris–HCl, 50 mM NaCl, pH 8.0 overnight at 4°C. Proteins were concentrated to 1 mg/ml prior to binding assays.

## ELISA with biotinylated glycans

For glycan ELISA, purified HA trimers were pre-complexed with anti-His mouse IgG (Invitrogen) and HRP-conjugated goat anti-mouse IgG (Pierce) and then diluted in series to required assay concentrations (40–0.05 μg/ml final). Preparation of streptavidin-coated plates with biotinylated glycans, incubation, and washing of pre-complexed HA dilutions was as described previously (Chandrasekaran *et al*, 2008; Peng *et al*, 2017).

## Differential scanning calorimetry

Thermal denaturation was studied using a nano-DSC calorimeter (TA instruments, Etten-Leur, the Netherlands). HA proteins were eluted from the streptavidin bead in PBS with 2.5 mM desthiobiotin, and 100 μg of protein was tested. After loading the sample into the cell, thermal denaturation was probed at a scan rate of 60°C/h. Buffer correction, normalization, and baseline subtraction procedures were applied before the data were analyzed using NanoAnalyze Software v.3.3.0 (TA Instruments). The data were fitted using a non-two-state model.

## Ethical statement

The chicken tissues were obtained from the tissue archive of the Veterinary Pathologic Diagnostic Center, Faculty of Veterinary Medicine, Utrecht University, the Netherlands. This archive is composed

---

**The paper explained**

**Problem**

Zoonotic influenza A virus infections warrant thorough analyses of how human-type receptor specificity can be achieved as a known factor of pandemic risk.

**Results**

We identified G225D as a determinant that confers human-type receptor specificity to the zoonotic H6N1 HA. Specificity was assessed using glycan arrays and analyzed at the atomic level from the crystal structure. Receptor switching was confirmed by binding to human tracheal tissues and a glycan ELISA assay. Using differential scanning calorimetry, we show that the H6 G225D still maintains a low, avian-like thermostability profile.

**Impact**

This work highlights that avian H6N1 influenza A virus can acquire human-type receptor specificity with a single amino acid mutation that does not change specificity in other avian virus subtypes. Knowledge of specific amino acid mutations that confer human-type receptor specificity will add to current influenza surveillance efforts.

---

of paraffin blocks with tissues maintained for diagnostic purposes; no permission of the Committee on the Ethics of Animal Experiment is required.

Anonymized human tissues were obtained under Service Level Agreement from the University Medical Centre Utrecht, the Netherlands. Use of anonymous material for scientific purposes is part of the standard treatment contract with patients, and therefore, informed consent procedure was not required according to the institutional medical ethical review board.

## Accession numbers

Atomic coordinates and structure factors have been deposited in the Protein Data Bank (PDB) under accession codes 5T08 for Taiwan2 H6 G225D HA in apo form and 5T0B, 5T0E, and 5T0D in complex with 6′-SLN, LSTa, and 3′-SLN.

**Expanded View** for this article is available online.

## Acknowledgements

This work was funded in part by National Institutes of Health grants R56 AI117675 (to I.A.W) and R01 AI114730 (to J.C.P), the Netherlands Organization for Scientific Research (NWO) VENI (R.P.D.V) and MEERVOUD (M.H.V.) grants and Rubicon (R.P.D.V) fellowship, and the Kwang Hua Educational Foundation (J.C.P). R.W.S. is a recipient of a Vidi grant from the Netherlands Organization for Scientific Research (NWO) and a Starting Investigator Grant from the European Research Council (ERC-StG-2011–280829-SHEV). A.J.T. is the recipient of an EMBO Long-term Fellowship (EMBO ALTF 963-2014). Several glycans used for HA binding assays were provided by the Consortium for Functional Glycomics (http://www.functionalglycomics.org/) funded by NIGMS grant GM62116 (J.C.P.). We thank R. Stanfield, X. Dai, and M. Elsliger for crystallographic and computational support; Henry Tien of the Robotics Core at the Joint Center for Structural Genomics and the Wilson laboratory for automated crystal screening; and the staff at the Advanced Photon Source beamline 23ID-B (GM/CA CAT) and the Stanford Synchrotron Radiation Lightsource (SSRL)

beamlines 11-1 and 12-2. GM/CA CAT is funded in whole or in part with federal funds from the National Cancer Institute (Y1-CO-1020) and NIGMS (Y1-GM-1104). Use of the Advanced Photon Source was supported by the U.S. Department of Energy (DOE), Basic Energy Sciences, Office of Science, under contract no. DE-AC02-06CH11357. The SSRL is a Directorate of Stanford Linear Accelerator Center National Accelerator Laboratory and an Office of Science User Facility operated for the U.S. DOE Office of Science by Stanford University. The SSRL Structural Molecular Biology Program is supported by the DOE Office of Biological and Environmental Research and by the NIH, NIGMS (including P41GM103393), and the National Center for Research Resources (NCRR, P41RR001209). This is manuscript 29406 from The Scripps Research Institute.

## Author contributions

Project design by RPV, NT, WP, IAW, and JCP; glycan array studies by RPV and RM; tissue staining studies by RPV, INAW, KMB, and MHV; X-ray structure determination, protein production, and analysis by NT, WY, and XZ; glycan ELISA by AJT; DSC measurements by ATTP, MJB, and RWS, and manuscript written by RPV, NT, AJT, IAW, and JCP.

## Conflict of interest

The authors declare that they have no conflict of interest.

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
