## [Review Process File · EMBO Molecular Medicine]

A single mutation in Taiwanese H6N1 influenza hemagglutinin switches binding to human-type receptors

Robert P de Vries, Netanel Tzarum, Wenjie Peng, Andrew J Thompson, Iresha N Ambepitiya Wickramasinghe, Alba T Torrents de la Pena, Marielle J van Breemen, Kim M Bouwman, Xueyong Zhu, Ryan McBride, Wenli Yu, Rogier W Sanders, Monique H. Verheije, Ian A Wilson

Corresponding author: Ian Wilson & James Paulson, The Scripps Research Institute

Review timeline:

Submission date:	20 February 2017
Editorial Decision:	16 February 2017
Additional Correspondence:	18 February 2017
Editorial Decision:	28 February 2017
Revision received:	10 May 2017
Editorial Decision:	31 May 2017
Revision received:	09 June 2017
Accepted:	12 June 2017

Transaction Report:

Editor: Céline Carret

1st Editorial Decision

16 February 2017

Thank you for submitting your manuscript for consideration by the EMBO Journal. It has now been seen by four referees whose comments are shown below.

As you will see, referee #1 evaluated DURC of your study. Referees 2-4 acknowledge that your work is important to evaluate the potential of H6 viruses to infect humans, and they provide constructive input on currently missing controls and on how to further strengthen your study. However, it is also clear from the report of referee #4 that a broader conceptual advance is not given. Given this input, and as we are a general interest journal, we discussed your manuscript again within our editorial team, and we sought further advice from another expert and trusted advisor of our journal. I am afraid that this expert also did not think that the conceptual advance provided is sufficient for publication here. I am thus afraid we concluded that we cannot offer to publish your manuscript in The EMBO Journal. I am very sorry that I couldn't bring better news this time.

This being said, we of course noted the support from referees 2 and 3, especially with regard to the pandemic potential of a mutated H6N1. I therefore took the liberty to discuss your work with my colleague Dr. Céline Carret, editor at our sister journal EMBO Molecular Medicine (EMM; embomolmed.org). EMM has a much more clinical/translational angle, and I am happy to say that Céline finds your study interesting from that point of view. Céline would be willing to seek arbitrating advice on your work to get further input on whether it is suitable for publication in EMM upon addressing the points made by the referees, and whether the further insight into pandemic

potential of the mutated HA as requested by referee #4 is needed.
I hope you view this positively. Should you request any additional info, please contact Céline directly. Her email address is: c.carret@embomolmed.org

REFEREE REPORTS

Referee #1:

Referring to the U.S. Government Policy for Institutional Oversight of Life Sciences Dual Use Research of Concern (DURC), according to its section 6.2.1., the project that is here being considered must be evaluated for its DURC potential because its researchers work with a strain of highly pathogenic influenza virus.

Having come to the conclusion stated above, the questions that must be answered are, which categories of experiments listed in section 6.2.2. are of concern? To this reviewer, there are two categories that might be of concern in regards to this project. The first reads "a), Enhances the harmful consequences of the agent;" and the second reads, "f) Enhances the susceptibility of a host population to the agent."

The researchers found that "...the H6 HA has the capacity to completely switch to human-type receptor specificity with a single nucleotide change..." There are three other aspects to the research. First, "...the mutant H6 was not place into a replication-competent virus and studies were restricted to in vitro analysis of a single viral protein." Second, "...it is well known that an influenza polymerase E627K mutation is also required for optimal infection; however, E627K is not present in avian and human H6N1 isolates." Third, "...other as yet to be identified properties many need to be optimized for adaption of the virus for transmission in man."

Based on what is revealed by the researchers, it is my belief that should the mutant H6N1 strain escape the laboratory, there will be no harmful consequences of the agent, nor will the studied virus be likely to cause humans to be more susceptible to it than its wild progenitor (which has caused only one human infection). For these reasons I do not believe that the findings and results of this project have DURC potential.

As an aside, though this project is basic research, I think its findings would be very important to public health was the H6N1 virus to shift and become infectious and/or transmissible to humans.

Referee #2:

In this manuscript, the authors examined the effect of HA mutations that confer human-type receptor recognition in H1, H2, and H3 viruses on the receptor-binding properties of H6HA protein by using a glycan microarray. On the basis of the glycan array data, they focused on the H6HA-G225D mutation. They analyzed the binding of HA-G225D mutant proteins to human and chicken tracheal tissue sections and determined the X-ray structures of H6HA-G225D mutant protein in complex with human and avian receptor analogs.

They found that introduction of the HA-G225D mutation into H6HA of A/Taiwan/2/2013 completely switch the binding of specificity from avian-type receptors (NeuAc 2,3Gal) to human-type receptors (NeuAc 2,6Gal) and led to binding to human trachea. Their crystal structure analysis showed that HA-225D forms hydrogen bonds with a human-type receptor analog, 6'-SLN, and that the angle between the Sia and Gal of 6'-SLN is similar to the interaction of human analogs with HAs of H1N1pdm 2009 and H2N2 viruses. The authors concluded that the single amino acid mutation G225D in the HA of H6N1 enables the virus to interact with human-type receptors.

The findings described in this study are important for our fundamental understanding of H6 HA recognition of human-type receptors and are useful to evaluate the potential of H6 viruses to infect humans. However, the authors should address the following points for clarification:

1. Although the authors concluded that the HA-G225D mutation alone can switch the receptor-binding specificity of H6N1 viruses from avian-receptor recognition to human-receptor recognition, A/Taiwan/2/2013 possess a unique mutation, P186L, that is not present in other H6N1 viruses.

Based on the data from Fig. 2G, the single HA-G225D mutation might not confer significant binding to human receptors to most H6 viruses that possess HA-186P. Therefore, they authors should assess the effect of the HA-G225D mutation in combination with the effect of P186L.

2. In Fig. 4 and Fig. S4. By using the data from the crystal structure analysis, the authors should provide more structurally detailed observations that explain the mechanism responsible for HA-G225 protein binding to the human-type receptor. For example, the authors should explain how the bond formation between the human receptor analog and the HA molecule could be changed by the G225D mutation.

3. The authors should show which numbering system was used for the amino acid positions of the HA protein.

4. In the experiments using human and chicken tissue sections, the authors should show the distribution of human and avian receptors on the chicken trachea used in this study.

5. Page 6, lines 9 and 10. Is it correct to say 'a2-3 linked (#9)'? The glycan #9 is a non-sialylated control (Table S1).

6. Page 8, Line 10. "Fig. 2D" should be "Fig. 4D".

7. Page 11, line 2. "is" should be "it".

8. Figure 1, A. What do '+' and '-' indicate?

9. Page 12, line 7. What is reference (7)?

Referee #3:

This is an interesting manuscript analysing the receptor binding characteristics of hemagglutinin (HA) from an H6N1 influenza virus to avian and human receptor analogues and presents important results. The manuscript shows that by introduction of a substitution G225D of HA1 in the H6 HA, expressed in 293T cells, the receptor binding finger-print changes from one in which receptor analogues of the avian receptor for influenza viruses predominates to one in which receptor analogues of the human receptor predominate. That is the virus receptor binding profile changes from binding sialic acid linked in a 2-3 linkage to the adjacent sugar to one in which the linkage is 2-6. The authors also determine the structure of the mutant and wild-type HAs both either in a complex with human or various avian receptor analogues or in an uncomplexed form.

Results like these are used as indications of the zoonotic or pandemic threat of the H6N1 virus and therefore have a practical consequence.

Overall the results look very interesting, but the authors do not comprehensively describe all the results obtained in the Results section of the manuscript - rather in the text they focus on drawing conclusions solely on residue 225. I think that a more comprehensive description of the results could be beneficial. Examples include: the Q226L substitution changing the receptor binding profile to one recognising 2-3 linked sulfated sugars, an intermediate profile in the V190D, mutant with some extension of that profile in the V190D, G225D double-mutant (all figure 1). Also the results derived from the substitutions made at residues 186 could also be fully presented, as well as those at residues 222 of HA1 and 227 (figure 2). Although some of these results are described in the figure legends in my opinion they would be better placed in the text of the Results.

It is worth noting that in Figure 1 results from an HA synthesised from a 2010 H3N2 sequence is presented. The HA gene sequence of A/Hong Kong/6934/2010 does not encode the amino acid sequences shown for the selected residues in Figure 1A for H3N2 viruses - those shown are a consensus of older H3N2 viruses. The residues, as determined on the MDCK-siat1 propagated virus, are at HA1 positions 190D, 225N, 226I and 228S. These sequences being typical of viruses collected between 2005 and 2013. Perhaps the authors might have considered presenting in their

results the receptor binding profile of a pre-2000 H3N2 virus. In addition it is surprising there is little reference overall to the role in of residues 222 and 225 in recent (post-2000) H3N2 viruses in the context of the binding observed for the H6 HA to the human receptor.

Minor points.

Page 2 Line 13. The authors cannot state that the G225D HA "specifically binds to human trachea epithelium" as epithelia from other species, apart from chicken, have not been tested. This should be changed to "is able to bind to human trachea epithelium", ie using similar wording to that used by the authors on Page 7 Line 14.

Page 5 Line 17. Is de Vries et al (2013) might not be the most appropriate reference for here as it merely refers to an earlier reference. Suggested is that the authors use de Vries et al (2010) Virology 17-25 as this is more descriptive.

Page 7 Line 8. The use of the word 'coincide' with regard to binding to the human respiratory epithelium and transmission might be considered as an overstatement. I suggest that this sentence be modified to tone down the coincidence since other factors are also known to be important for transmission of viruses.

Page 8 line 6. I suggest the phi angle be indicated in Figure S3.

Page 8 line 10. I think the authors are referring to Figure 4D. Again the phi angle could be usefully shown.

Page 8 line 19. I think that the authors need to precisely check which panels of Figure S4 are referred to. They might want to rewrite this paragraph.

Page 10 lines 14 and 15. The authors describe the binding of the 225G HA to the avian epithelium as strong, but there is no measure of affinity presented for this assay of association with the avian or human epithelium.

Page 12 Materials and Methods Lines 3 to 9. The names of the virus HA genes that were expressed does not match completely the genes used in the paper.

The authors should be aware that in the materials and methods section jargon is used in some parts and some usage of English could certainly be improved in these parts. Editing is encouraged.

Referee #4:

Vries et al describe the finding that in the Taiwanese poultry H6N1 virus lineage, which has resulted in a human infection, mutation of position 225 to an aspartic acid (D) leads to the phenotype of reduced binding to α 2-3 linked sialic acid and increased binding to α 2-6 linked sialic acid. The authors use recombinant HA proteins and glycan arrays plus ex-vivo tracheal sections to probe the attachment specificity of this mutation.

The science presented in this manuscript has been performed soundly and the results are presented clearly. The standard of writing is good and needs little editing. However the information offered does little to offer increased knowledge to the field of influenza virus host range barriers because of the techniques employed to analyse this question. This may be because of the US moratorium but it stands in the way of more convincing and relevant data.

To improve the clarity and impact of the manuscript the following points could be addressed:

1. The authors include both an avian virus HA control (H5) and human virus HA (H3) on the glycan array alongside the H6 and subsequent H6 mutants. The usefulness of these controls can be improved by allowing the readers to align common glycans on the array, currently it is unclear if they are binding the same hits to the same degree.
2. Indeed it is hard to determine the avidity of the mutants or wt HA protein binding to the glycans and what relevance this actually has in the context of the different hosts infection. For example the 225D mutation has shifted the type of glycan it binds to from α 2-3 linked SA to α 2-6 SA but what is the affinity of the α 2-6 binding in comparison to the H3N2 human strain that the authors have

utilised alongside. It would improve the manuscript to have these direct comparisons.

3. The binding to the ex vivo tracheal sections is convincing however, the extent of binding of the G225D does look to be less than for the H3 human HA. Does this again reflect a quantitative difference that is just not measured in the glycan arrays? Moreover, no attempt has been made to demonstrate what the HA proteins are binding to in these sections. Accompanying lectin stains indicating the presence of a2-3 or a2-6 sialic acid on the surface would be more convincing and/or loss of binding after sialidase treatment. In addition it is not stated in the manuscript if only a single section was probed by the various HA proteins or whether the shown staining is representative of multiple sections from a single host or ideally representative of multiple individuals.

4. To really improve the impact of this manuscript comparison of the relevance of the HA protein mutations on infectivity of tracheal cultures in the context of a whole virus, even if that virus was a PR8 backbone recombinant possessing the HA and NA of the H6N1 or control virus strains. Some understanding of the relationship between what the HA protein attachment assays are demonstrating and real replicative ability in different hosts. It is not known for example if the H6 HA mutant 225D can even be rescued into the context of a viable virus and important question which adds to the relevance of the data displayed in this manuscript.

5. Correctly in the discussion the authors acknowledge that other genetic changes might be required to transform H6N1 virus into a human transmitting virus. Throughout the paper they fail to acknowledge the growing idea that pH stability of the HA protein is important in this respect and might be adversely affected by receptor binding changes. In the absence of rescue of infectious virus are there biophysical measures the authors can conduct that might assess protein stability at different pH? There are also clever genetic ways of making a non-infectious virus that does bear a mutated HA by pseudotyping in cells that express the altered HA.

6. The authors imply that others have been confused about the sialic acid specificity of H6 HA because the use inappropriate glycans. The glycan to which G225D H6 HA binds are the long chain glycans that are also preferred by recent H3 HA viruses. However the crystal structure does not reveal why such long glycans are preferred since only short glycans are cocrystallized. It is not clear what the long chains add and why this makes such a difference.

To summarize the information in the manuscript might be seen by many noninfluenza experts as incremental, just one more HA subtype looked at and one more mutation. If the big message is that one single amino acid change is enough for H6 to cause a pandemic then Show in at least one quantitative assay that the mutant H6 HA is really as 'good' at binding the relevant receptors as an H3 human HA.

Find a way to assess the effects of this mutation in a more relevant viral like context

Assess and discuss whether pH of fusion of HA is affected by this change, and what further mutations might be required to support adaptation .

Additional Correspondence

18 February 2017

As you will see below, Andrea Leibfried mentioned that she had talked to you about our manuscript demonstrating that the avian H6N1 virus that had infected a Taiwanese woman can acquire human type receptor specificity with a single nucleotide change. Since this phenotype is widely believed to be required for transmission in humans, we feel it is important to the surveillance of avian influenza viruses, particularly since another wave of H7N9 influenza viruses is occurring in China.

We were happy to here that you felt the manuscript would be a good fit for EMBO Molecular Medicine, and would like to inquire what would be the next step to transfer the manuscript to your journal. As you will see I have appended the decision letter from Andrea, and have no problem for the reviews to be a part of your evaluation of the MS.

1st Editorial Decision

28 February 2017

Thank you for the submission of your manuscript to EMBO Molecular Medicine and for your patience while we were seeking additional input from an external expert in the field.

I have now heard from this advisor and am happy to report that if you could answer most of the

concerns raised by the referees (perhaps not all of referee 4, but the data on the relative affinities for ex.), we would be willing to consider your article for publication in EMBO Molecular Medicine, pending another round of review. We believe, like our advisor, that a revision in the line suggested by referees would greatly improve the study and can be done without the need of making a recombinant virus.

Revised manuscripts should be submitted within three months of a request for revision; they will otherwise be treated as new submissions, except under exceptional circumstances in which a short extension is obtained from the editor.

Please note that it is EMBO Molecular Medicine policy to allow only a single round of revision and that, as acceptance or rejection of the manuscript will depend on another round of review, your responses should be as complete as possible.

I look forward to seeing a revised form of your manuscript as soon as possible.

1st Revision - authors' response

10 May 2017

Responses to the Reviewers Comments:

Referee #1:

Referring to the U.S. Government Policy for Institutional Oversight of Life Sciences Dual Use Research of Concern (DURC), according to its section 6.2.1., the project that is here being considered must be evaluated for its DURC potential because its researchers work with a strain of highly pathogenic influenza virus.

Having come to the conclusion stated above, the questions that must be answered are, which categories of experiments listed in section 6.2.2. are of concern? To this reviewer, there are two categories that might be of concern in regards to this project. The first reads "a), Enhances the harmful consequences of the agent;" and the second reads, "f) Enhances the susceptibility of a host population to the agent."

The researchers found that "...the H6 HA has the capacity to completely switch to human-type receptor specificity with a single nucleotide change..." There are three other aspects to the research. First, "...the mutant H6 was not place into a replication-competent virus and studies were restricted to in vitro analysis of a single viral protein." Second, "...it is well known that an influenza polymerase E627K mutation is also required for optimal infection; however, E627K is not present in avian and human H6N1 isolates." Third, "...other as yet to be identified properties many need to be optimized for adaption of the virus for transmission in man."

Based on what is revealed by the researchers, it is my belief that should the mutant H6N1 strain escape the laboratory, there will be no harmful consequences of the agent, nor will the studied virus be likely to cause humans to be more susceptible to it than its wild progenitor (which has caused only one human infection). For these reasons I do not believe that the findings and results of this project have DURC potential.

As an aside, though this project is basic research, I think its findings would be very important to public health was the H6N1 virus to shift and become infectious and/or transmissible to humans.

We thank the reviewer for the positive comments about the work, and conclusion that the findings presented do not have DURC potential. As discussed below, several referees have suggested experiments that would be gain of function experiments with viruses containing the mutations we describe. However, we have been advised by program officials at NIH that this work would fall under the moratorium.

Referee #2:

In this manuscript, the authors examined the effect of HA mutations that confer human-type receptor recognition in H1, H2, and H3 viruses on the receptor-binding properties of H6HA protein by using a glycan microarray. On the basis of the glycan array data, they focused on the H6HA-G225D mutation. They analyzed the binding of HA-G225D mutant proteins to human and chicken tracheal tissue sections and determined the X-ray structures of H6HA-G225D mutant protein in complex with human and avian receptor analogs.

They found that introduction of the HA-G225D mutation into H6HA of A/Taiwan/2/2013 completely switch the binding of specificity from avian-type receptors (NeuAc α 2,3Gal) to human-type receptors (NeuAc α 2,6Gal) and led to binding to human trachea. Their crystal structure analysis showed that HA-225D forms hydrogen bonds with a human-type receptor analog, 6'-SLN, and that the angle between the Sia and Gal of 6'-SLN is similar to the interaction of human analogs with HAs of H1N1pdm 2009 and H2N2 viruses. The authors concluded that the single amino acid mutation G225D in the HA of H6N1 enables the virus to interact with human-type receptors. The findings described in this study are important for our fundamental understanding of H6 HA recognition of human-type receptors and are useful to evaluate the potential of H6 viruses to infect humans. However, the authors should address the following points for clarification:

We thank the reviewer for the positive comments

1. Although the authors concluded that the HA-G225D mutation alone can switch the receptor-binding specificity of H6N1 viruses from avian-receptor recognition to human-receptor recognition, A/Taiwan/2/2013 possess a unique mutation, P186L, that is not present in other H6N1 viruses. Based on the data from Fig. 2G, the single HA-G225D mutation might not confer significant binding to human receptors to most H6 viruses that possess HA-186P. Therefore, they authors should assess the effect of the HA-G225D mutation in combination with the effect of P186L.

We entirely agree that the G225D mutation converts receptor specificity to human-type in the context of the P186L mutation that occurs in the human A/Taiwan/2/2013 isolate. We have presented the data suggested by the referee in Figure 2, showing that reverting L186 in this human isolate back to P186 (L186P) in the avian isolates exhibits avian-type specificity, and adding G225D shows binding to both and avian- human-type receptors. Thus, G225D also confers binding to human type receptors with P186, but the impact on switch to human type receptor specificity is more complete with L186 in the human H6 isolate. This entire section has been rewritten to improve flow and highlight this difference in A/Taiwan/2/2013 as suggested by the referee.

2. In Fig. 4 and Fig. S4. By using the data from the crystal structure analysis, the authors should provide more structurally detailed observations that explain the mechanism responsible for HA-G225 protein binding to the human-type receptor. For example, the authors should explain how the bond formation between the human receptor analog and the HA molecule could be changed by the G225D mutation.

We agree and now describe the interactions that the 6'SLN forms with the 225-side chain that stabilize the cis conformation.

3. The authors should show which numbering system was used for the amino acid positions of the HA protein.

We now indicate that we use H3 numbering.

4. In the experiments using human and chicken tissue sections, the authors should show the distribution of human and avian receptors on the chicken trachea used in this study.

We added a figure in the supplementary data (Fig S2) in which we use commonly used plant lectins to detect α 2-3 and α 2-6 sialosides in tissue sections. The results show that chicken tissues predominantly contain avian-type receptors using MAA lectins and that human trachea predominantly expresses human-type receptors as detected by SNA lection. All staining was sialic acid dependent since treatment with a bacterial neuraminidase eliminated binding by the lectins

5. Page 6, lines 9 and 10. Is it correct to say ' α 2-3 linked (#9)'? The glycan #9 is a non-sialylated control (Table S1).

Thank you, this is indeed a typo and is corrected

6. Page 8, Line 10. "Fig. 2D" should be "Fig. 4D".

Corrected

7. Page 11, line 2. "is" should be "it".

Corrected

8. Figure 1, A. What do '+' and '-' indicate?

We added a line in the figure legends indicating that the + or – refers to binding of either human or avian-type receptors.

9. Page 12, line 7. What is reference (7)?

This is corrected

Referee #3:

This is an interesting manuscript analysing the receptor binding characteristics of hemagglutinin (HA) from an H6N1 influenza virus to avian and human receptor analogues and presents important results. The manuscript shows that by introduction of a substitution G225D of HA1 in the H6 HA, expressed in 293T cells, the receptor binding finger-print changes from one in which receptor analogues of the avian receptor for influenza viruses predominates to one in which receptor analogues of the human receptor predominate. That is the virus receptor binding profile changes from binding sialic acid linked in a 2-3 linkage to the adjacent sugar to one in which the linkage is 2-6. The authors also determine the structure of the mutant and wild-type HAs both either in a complex with human or various avian receptor analogues or in an uncomplexed form.

Results like these are used as indications of the zoonotic or pandemic threat of the H6N1 virus and therefore have a practical consequence.

We thank the referee for the positive view.

Overall the results look very interesting, but the authors do not comprehensively describe all the results obtained in the Results section of the manuscript - rather in the text they focus on drawing conclusions solely on residue 225. I think that a more comprehensive description of the results could be beneficial. Examples include: the Q226L substitution changing the receptor binding profile to one recognising 2-3 linked sulfated sugars, an intermediate profile in the V190D, mutant with some extension of that profile in the V190D, G225D double-mutant (all figure 1). Also the results derived from the substitutions made at residues 186 could also be fully presented, as well as those at residues 222 of HA1 and 227 (figure 2). Although some of these results are described in the figure legends in my opinion they would be better placed in the text of the Results.

We thank the reviewer for this suggestion; as mentioned above for referee 2, we have made a concerted effort to improve the flow and clarity of the sections describing the receptor specificity of the different H6 mutants.

It is worth noting that in Figure 1 results from an HA synthesised from a 2010 H3N2 sequence is presented. The HA gene sequence of A/Hong Kong/6934/2010 does not encode the amino acid sequences shown for the selected residues in Figure 1A for H3N2 viruses - those shown are a consensus of older H3N2 viruses. The residues, as determined on the MDCK-siat1 propagated virus, are at HA1 positions 190D, 225N, 226I and 228S. These sequences being typical of viruses collected between 2005 and 2013. Perhaps the authors might have considered presenting in their results the receptor binding profile of a pre-2000 H3N2 virus. In addition it is surprising there is little reference overall to the role in of residues 222 and 225 in recent (post-2000) H3N2 viruses in the context of the binding observed for the H6 HA to the human receptor.

The reviewer makes an excellent point that the sequence of the control H3 human virus shown in Figure 1 panel A was a consensus including older H3 representatives. Contemporary (post-2000) H3N2 viruses, including A/Hong Kong/6934/2010, have evolved a different repertoire of amino acids at the highlighted positions, while still maintaining human-type receptor specificity. We recently published an extensive analysis of receptor specificity of H3N2 viruses from 1968-2011, and find on our glycan array that receptor specificity is conserved for a subset of human-type receptor glycans with extended chains.

We have now modified Fig 1A and the introduction to reflect the fact that these canonical residues changed in H3 viruses after 2000, and included H3 in our analysis of positions 226 and 228. We also added two references that show that recent H3N2 viruses exhibit reduced avidity to short human-type receptor glycans in laboratory assays, but did not change receptor binding specificity to more extended human-type glycans during evolution.

Minor points.

Page 2 Line 13. The authors cannot state that the G225D HA "specifically binds to human trachea epithelium" as epithelia from other species, apart from chicken, have not been tested. This should be changed to "is able to bind to human trachea epithelium", ie using similar wording to that used by the authors on Page 7 Line 14.

Corrected, thank you.

Page 5 Line 17. Is de Vries et al (2013) might not be the most appropriate reference for here as it merely refers to an earlier reference. Suggested is that the authors use de Vries et al (2010) Virology 17-25 as this is more descriptive.

Corrected, thank you.

Page 7 Line 8. The use of the word 'coincide' with regard to binding to the human respiratory epithelium and transmission might be considered as an overstatement. I suggest that this sentence be modified to tone down the coincidence since other factors are also known to be important for transmission of viruses.

We adjusted this sentence, and now say that binding to human respiratory epithelium is an important factor.

Page 8 line 6. I suggest the phi angle be indicated in Figure S3.

Thank you we added this in Fig. S3, which is now Fig. S4.

Page 8 line 10. I think the authors are referring to Figure 4D. Again the phi angle could be usefully shown.

Thank you we added this.

Page 8 line 19. I think that the authors need to precisely check which panels of Figure S4 are referred to. They might want to rewrite this paragraph.

Thank you we rewrote this paragraph and we clarified the reference to the figure in the main text.

Page 10 lines 14 and 15. The authors describe the binding of the 225G HA to the avian epithelium as strong, but there is no measure of affinity presented for this assay of association with the avian or human epithelium.

The reviewer is right as these assays are qualitative. We removed “strong” in our wording.

Page 12 Materials and Methods Lines 3 to 9. The names of the virus HA genes that were expressed does not match completely the genes used in the paper.

The authors should be aware that in the materials and methods section jargon is used in some parts and some usage of English could certainly be improved in these parts. Editing is encouraged.

Thank you, inclusion of A/Kentucky/07 as the human control virus in Materials and Methods was a typographical error. All experiments were performed using H3 HA from A/Hong Kong/6934/2010 (H3N2), as detailed throughout the manuscript. This has now been corrected Thank you for the suggestion; we have thoroughly checked the manuscript including materials and methods for jargon and English.

Referee #4:

Vries et al describe the finding that in the Taiwanese poultry H6N1 virus lineage, which has resulted in a human infection, mutation of position 225 to an aspartic acid (D) leads to the phenotype of reduced binding to α 2-3 linked sialic acid and increased binding to α 2-6 linked sialic acid. The authors use recombinant HA proteins and glycan arrays plus ex-vivo tracheal sections to probe the attachment specificity of this mutation.

The science presented in this manuscript has been performed soundly and the results are presented clearly. The standard of writing is good and needs little editing. However the information offered does little to offer increased knowledge to the field of influenza virus host range barriers because of the techniques employed to analyse this question. This may be because of the US moratorium but it stands in the way of more convincing and relevant data.

We first would like to thank the reviewer for the positive words on the sound science in our manuscript. We are indeed not allowed to create a virus with mutant avian gene that would provide gain-of-function, even in a viral plasmid to create a virus with a PR8 backbone containing the H6N1 HA and NA genes. Ultimately, this may be possible, but the moratorium is still in place, and there is as yet no defined review mechanism to submit proposals for such research. It is likely, however, that publication of this manuscript might possibly result in such experiments being done, but most likely in Europe or China, where there are fewer restrictions imposed for gain-of-function research than in the United States.

To improve the clarity and impact of the manuscript the following points could be addressed:

1. The authors include both an avian virus HA control (H5) and human virus HA (H3) on the glycan array alongside the H6 and subsequent H6 mutants. The usefulness of these controls can be improved by allowing the readers to align common glycans on the array, currently it is unclear if they are binding the same hits to the same degree.

We agree with the reviewer that a table would increase clarity; we therefore added a table to the supplemental information (Table S3).

2. Indeed it is hard to determine the avidity of the mutants or wt HA protein binding to the glycans and what relevance this actually has in the context of the different hosts infection. For example the 225D mutation has shifted the type of glycan it binds to from α 2-3 linked SA to α 2-6 SA but what is

the affinity of the α 2-6 binding in comparison to the H3N2 human strain that the authors have utilised alongside. It would improve the manuscript to have these direct comparisons.

The reviewer makes an excellent point regarding the lack of an avidity assay. We have now added an avidity assay using biotinylated N-glycans with different lengths of LacNAc repeats terminating with either avian or human-type receptors. The results show that the G225D mutant lost all binding to avian-type receptors and now specifically binds human-type receptors (Fig 5). The avidities are comparable to those for human H3N2 viruses that we have recently published (Peng et al., 2017).

3. The binding to the ex vivo tracheal sections is convincing however, the extent of binding of the G225D does look to be less than for the H3 human HA. Does this again reflect a quantitative difference that is just not measured in the glycan arrays? Moreover, no attempt has been made to demonstrate what the HA proteins are binding to in these sections. Accompanying lectin stains indicating the presence of α 2-3 or α 2-6 sialic acid on the surface would be more convincing and/or loss of binding after sialidase treatment.

We agree and have now included plant lectin control staining in the supplemental information (Figure S2). The reviewer is also right that this reflects a quantitative difference that we are not able to pick up in the glycan arrays. The new ELISA assay included in Figure 5 does demonstrate this low avidity.

In addition it is not stated in the manuscript if only a single section was probed by the various HA proteins or whether the shown staining is representative of multiple sections from a single host or ideally representative of multiple individuals.

We have now added in the figure legends that the staining is a representative of three independent assays. We have also repeated the experiments on tracheal section of different individuals and chickens with similar results.

4. To really improve the impact of this manuscript comparison of the relevance of the HA protein mutations on infectivity of tracheal cultures in the context of a whole virus, even if that virus was a PR8 backbone recombinant possessing the HA and NA of the H6N1 or control virus strains. Some understanding of the relationship between what the HA protein attachment assays are demonstrating and real replicative ability in different hosts. It is not known for example if the H6 HA mutant 225D can even be rescued into the context of a viable virus and important question, which adds to the relevance of the data displayed in this manuscript.

As mentioned above, as much as we would like to perform this experiment, at this time it is unclear when we might receive permission to perform such research.

5. Correctly in the discussion the authors acknowledge that other genetic changes might be required to transform H6N1 virus into a human transmitting virus. Throughout the paper they fail to acknowledge the growing idea that pH stability of the HA protein is important in this respect and might be adversely affected by receptor binding changes. In the absence of rescue of infectious virus are there biophysical measures the authors can conduct that might assess protein stability at different pH? There are also clever genetic ways of making a non-infectious virus that does bear a mutated HA by pseudotyping in cells that express the altered HA.

We would like to thank the reviewer for bringing this up as indeed HA stability is an important feature for a zoonotic influenza virus to become transmissible between humans. We decided to analyze HA stability by Differential Scanning Calorimetry (DSC), in which we measure thermostability that has been directly correlated in the literature with pH stability. Indeed, the H6 wild type as well as the G225D mutant displays low thermostability comparable to H5 viruses that are not able to transmit between ferrets. As a human control, we used H1N1 A/CA/04/09 that indeed has higher thermostability. We included these data in new figure 6 and changed the manuscript accordingly.

6. The authors imply that others have been confused about the sialic acid specificity of H6 HA because the use inappropriate glycans. The glycan to which G225D H6 HA binds are the long chain

glycans that are also preferred by recent H3 HA viruses. However the crystal structure does not reveal why such long glycans are preferred since only short glycans are co-crystallized. It is not clear what the long chains add and why this makes such a difference.

The actual specificity of the binding of the key glycans is unchanged but the binding affinity is much decreased for shorter glycans that can be overcome by adding a large molar excess of ligands in the crystallization experiment. We believe that the longer chain bi-antennary glycans overcome the low affinity for monovalent binding to shorter glycans by enabling simultaneous binding to two protomers in the same HA trimer. A sense of this can be seen in the new data obtained with biantennary glycans of different lengths, where a jump in avidity is observed when with at least three LacNAc repeats. We have elaborated on this phenomenon in our recent paper (Peng et al., 2017) where we also showed from computational modeling that these glycans can engage two protomers within a single HA trimer at the same time (Peng et al., 2017). In the description of new Figure 5, we also now emphasize the similarity of the receptor binding profile of H6 G225D with that of human H3N2 viruses.

To summarize the information in the manuscript might be seen by many non-influenza experts as incremental, just one more HA subtype looked at and one more mutation. If the big message is that one single amino acid change is enough for H6 to cause a pandemic then:

-Show in at least one quantitative assay that the mutant H6 HA is really as 'good' at binding the relevant receptors as an H3 human HA.

We now included a glycan ELISA in which we show that H6 G225D indeed binds human-type receptors specifically, and with comparable avidity to human H3N2 viruses as described in Peng et al (2017).

-Find a way to assess the effects of this mutation in a more relevant viral like context

In lieu of virus experiments, we have included additional biological data with the avidity assays and thermostability of the HA.

-Assess and discuss whether pH of fusion of HA is affected by this change, and what further mutations might be required to support adaptation.

As mentioned above, we appreciate this suggestion. We have now analyzed the thermostability of the H6 proteins and show that their thermostability (that has been documented to correlate with the pH stability) is similar to avian isolates, and are thus not likely to transmit between ferrets. We have now elaborated on this point both in the results and discussion.

2nd Editorial Decision

31 May 2017

Thank you for the submission of your revised manuscript to EMBO Molecular Medicine. We have now received the enclosed reports from the referees that were asked to re-assess it. As you will see the reviewers are now globally supportive and I am pleased to inform you that we will be able to accept your manuscript pending the following final amendments:

1) Please address the minor text changes commented by referee 2. Please provide a letter INCLUDING the reviewer's reports and your detailed responses to their comments (as Word file).

2) Please carefully check the authors guidelines for formatting your supplemental information: Expanded view and Appendix (see: <http://embomolmed.embopress.org/authorguide#expandedview>). Your Appendix file should be labeled within the pdf as "appendix" in the title page and Table Of Content as well.

3) Figures:

-callouts: figure 1D is called before 1C, could you please update the order, maybe in the figure?

-Figure 5B, D is called for, but figure 5 has no sub panels, could you please update? is it figure 6?

-Figure 6 is not called for at all

-make sure that all sub panels are called for where appropriate
-Appendix figure S5 is of low resolution, could you please replace the figure within the pdf?

4) In the main manuscript file:

- please provide a running title
- please add up to 5 keywords
- please include the Paper Explained at the end of the main article
- references listed at the end of the article: when more than 20 authors, please use "et al". Otherwise, list all contributing authors.
- in M&M, please include an ethics statement regarding the human and chicken material.
- statistics: please indicate how many times an experiment was performed and when appropriate whether p-values were calculated. Please provide in legends exact n= and exact p= values, not a range.

5) Synopsis: Thank you for providing a synopsis text and image.

As the text was too long, I shortened it, please confirm you're happy with it:

"To transmit between humans, zoonotic viruses need to acquire human-type receptor specificity. Analyzing the effects of different amino acid mutations in the receptor binding site revealed that H6 HA only requires a single amino acid mutation to specifically adapt to human-type receptor binding."

The image provided is too small. Please provide a novel .jpeg file with the resolution 550px X 250-400px (currently it is given at 274 X 212)

6) As part of the EMBO Publications transparent editorial process initiative (see our Editorial at <http://embomolmed.embopress.org/content/2/9/329>), EMBO Molecular Medicine will publish online a Review Process File (RPF) to accompany accepted manuscripts.

In the event of acceptance, this file will be published in conjunction with your paper and will include the anonymous referee reports, your point-by-point response and all pertinent correspondence relating to the manuscript, including the initial correspondence to The EMBO Journal. If you do NOT want this file to be published, please inform the editorial office at contact@embomolmed.org.

Please submit your revised manuscript within two weeks. I look forward to seeing a revised form of your manuscript as soon as possible.

***** Reviewer's comments *****

Referee #2 (Remarks):

The authors properly responded to my comments.

Referee #4 (Comments on Novelty/Model System):

ethical issues would relate only to the information as no infectious virus was generated

Referee #4 (Remarks):

The manuscript is much improved and I commend the authors for taking on board suggestions to this end.

Here are some final suggestions to further improve the clarity of the paper:

1. Page 4 line 2 suggest reword 'did not alter receptor specificity' to 'retain preference for'
2. Page 4 line 21. We all agree that glycan microarrays are not quantitative so I suggest remove the

word strong.

3. Page 5 lines 6-9. This section is not clearly worded and without introducing the role that stability of HA plays in supporting transmission, a non influenza expert might be lost here. I suggest rewording this section.

4. Page 6 line 14 the authors might want to consider work from Peacock et al on H9 viruses and their binding to sulphated glycans as a more recent addition to the findings reported by Gambarayan.

5. Page 7 line 2 suggest clarifying that these HAs did not show differences to the same HA generated in HEK cells otherwise the sentence reads as if the wild type and mutant HA showed similar specificities to each other if they were produced in insect cells.

6. Page 7 line 18 and figure 2 Make clear if leucine is not found in any other human or avian viruses regardless of subtype or just within the H6 group? To help, I suggest adding a row in the table figure 2A- H6N1 currently included is specifically the Taiwan/2/2013 human isolate, add below a row that shows that avian isolates of H6N1 have P at 186.

7. Page 12 line 10 pH of 5.8 is still high for a human isolate; most studies show pH1N1 2009 virus HA fusion pH to be 5.5.

8. Page 14 line 15 reword it si well known that an influenza polymerase mutation such as E627K...(pH1N1 2009 viruses do not have this one)

2nd Revision - authors' response

09 June 2017

Responses to the Reviewers Comments:

Referee #2 (Remarks):

The authors properly responded to my comments.

We thank the referee for the support.

Referee #4 (Comments on Novelty/Model System):

Ethical issues would relate only to the information as no infectious virus was generated

Thank you

Referee #4 (Remarks):

The manuscript is much improved and I commend the authors for taking on board suggestions to this end.

Thank you for the supportive comment.

Here are some final suggestions to further improve the clarity of the paper:

1. Page 4 line 2 suggest reword 'did not alter receptor specifitiy' to 'retain preference for'

Thank you, we reworded it.

2. Page 4 line 21. We all agree that glycan microarrays are not quantitative so I suggest remove the word strong.

Thank you, we removed strong.

3. Page 5 lines 6-9. This section is not clearly worded and without introducing the role that stability of HA plays in supporting transmission, a non influenza expert might be lost here. I suggest rewording this section.

Thank you, we added a sentence that increasing HA stability is an important factor in the adaptation process of avian viruses to humans.

4. Page 6 line 14 the authors might want to consider work from Peacock et al on H9 viruses and their binding to sulphated glycans as a more recent addition to the findings reported by Gambarayan.

We added the appropriate reference, thank you.

5. Page 7 line 2 suggest clarifying that these HAs did not show differences to the same HA generated in HEK cells otherwise the sentence reads as if the wild type and mutant HA showed similar specificities to each other if they were produced in insect cells.

We clarified the wording here as suggested.

6. Page 7 line 18 and figure 2 Make clear if leucine is not found in any other human or avian viruses regardless of subtype or just within the H6 group? To help, I suggest adding a row in the table figure 2A- H6N1 currently included is specifically the Taiwan/2/2013 human isolate, add below a row that shows that avian isolates of H6N1 have P at 186.

Thank you, we added a line in the table to indicate that all other H6N1 isolate contain a P at 186.

7. Page 12 line 10 pH of 5.8 is still high for a human isolate; most studies show pH1N1 2009 virus HA fusion pH to be 5.5.

Several studies, as referenced, show that early 2009 pandemic viruses have a pH of fusion ranging from 5.5 to 5.8. we added the peacock reference as suggested. The CA04 strain used here is in fact the very first pH1N1 2009 strain identified.

8. Page 14 line 15 reword it si well known that an influenza polymerase mutation such as E627K...(pH1N1 2009 viruses do not have this one)

Thank you we adapted the sentence to "it is well known that an influenza polymerase E627K mutation is also often required for optimal infection"

Corresponding Author Name: James C Paulson

Journal Submitted to: EMBO MOL MED

Manuscript Number: EMM-2017-07726-T